# Parallel Double Greedy Submodular Maximization

**Xinghao Pan**[1] **Stefanie Jegelka**[1] **Joseph Gonzalez**[1] **Joseph Bradley**[1] **Michael I. Jordan**[1,2]
[1]Department of Electrical Engineering and Computer Science, and [2]Department of Statistics
University of California, Berkeley, Berkeley, CA USA 94720
{xinghao,stefje,jegonzal,josephkb,jordan}@eecs.berkeley.edu

## Abstract

Many machine learning problems can be reduced to the maximization of sub-modular functions. Although well understood in the serial setting, the parallel maximization of submodular functions remains an open area of research with recent results [1] only addressing monotone functions. The optimal algorithm for maximizing the more general class of non-monotone submodular functions was introduced by Buchbinder et al. [2] and follows a strongly serial double-greedy logic and program analysis. In this work, we propose two methods to parallelize the double-greedy algorithm. The first, *coordination-free* approach emphasizes speed at the cost of a weaker approximation guarantee. The second, *concurrency control* approach guarantees a tight 1/2-approximation, at the quantifiable cost of additional coordination and reduced parallelism. As a consequence we explore the tradeoff space between guaranteed performance and objective optimality. We implement and evaluate both algorithms on multi-core hardware and billion edge graphs, demonstrating both the scalability and tradeoffs of each approach.

## 1 Introduction

Many important problems including sensor placement [3], image co-segmentation [4], MAP inference for determinantal point processes [5], influence maximization in social networks [6], and document summarization [7] may be expressed as the maximization of a submodular function. The submodular formulation enables the use of targeted algorithms [2, 8] that offer theoretical worst-case guarantees on the quality of the solution. For several maximization problems of *monotone* submodular functions (satisfying $F(A) \leq F(B)$ for all $A \subseteq B$), a simple greedy algorithm [8] achieves the optimal approximation factor of $1 - \frac{1}{e}$. The optimal result for the wider, important class of *non-monotone* functions — an approximation guarantee of $1/2$ — is much more recent, and achieved by a *double greedy* algorithm by Buchbinder et al. [2].

While theoretically optimal, in practice these algorithms do not scale to large real world problems, since the inherently serial nature of the algorithms poses a challenge to leveraging advances in parallel hardware. This limitation raises the question of parallel algorithms for submodular maximization that ideally preserve the theoretical bounds, or weaken them gracefully, in a quantifiable manner.

In this paper, we address the challenge of parallelization of greedy algorithms, in particular the double greedy algorithm, from the perspective of *parallel transaction processing systems*. This alternative perspective allows us to apply advances in database research ranging from fast coordination-free approaches with limited guarantees to sophisticated concurrency control techniques which ensure a direct correspondence between parallel and serial executions at the expense of increased coordination.

We develop two parallel algorithms for the maximization of non-monotone submodular functions that operate at different points along the coordination tradeoff curve. We propose CF-2g as a coordination-free algorithm and characterize the effect of reduced coordination on the approximation ratio. By bounding the possible outcomes of concurrent transactions we introduce the CC-2g algorithm which

guarantees serializable parallel execution and retains the optimality of the double greedy algorithm at the expense of increased coordination. The primary contributions of this paper are:

1. We propose two parallel algorithms for unconstrained non-monotone submodular maximization, which trade off parallelism and tight approximation guarantees.

2. We provide approximation guarantees for CF-2g and analytically bound the expected loss in objective value for set-cover with costs and max-cut as running examples.

3. We prove that CC-2g preserves the optimality of the serial double greedy algorithm and analytically bound the additional coordination overhead for covering with costs and max-cut.

4. We demonstrate empirically using two synthetic and four real datasets that our parallel algorithms perform well in terms of both speed and objective values.

The rest of the paper is organized as follows. Sec. 2 discusses the problem of submodular maximization and introduces the double greedy algorithm. Sec. 3 provides background on concurrency control mechanisms. We describe and provide intuition for our CF-2g and CC-2g algorithms in Sec. 4 and Sec. 5, and then analyze the algorithms both theoretically (Sec. 6) and empirically (Sec. 7).

## 2   Submodular Maximization

A set function $F : 2^V \to \mathbb{R}$ defined over subsets of a ground set $V$ is *submodular* if it satisfies *diminishing marginal returns*: for all $A \subseteq B \subseteq V$ and $e \notin B$, it holds that $F(A \cup \{e\}) - F(A) \geq F(B \cup \{e\}) - F(B)$. Throughout this paper, we will assume that $F$ is nonnegative and $F(\emptyset) = 0$. Submodular functions have emerged in areas such as game theory [9], graph theory [10], combinatorial optimization [11], and machine learning [12, 13]. Casting machine learning problems as submodular optimization enables the use of algorithms for submodular maximization [2, 8] that offer theoretical worst-case guarantees on the quality of the solution.

While those algorithms confer strong guarantees, their design is inherently serial, limiting their usability in large-scale problems. Recent work has addressed faster [14] and parallel [1, 15, 16] versions of the greedy algorithm by Nemhauser et al. [8] for maximizing *monotone* submodular functions that satisfy $F(A) \leq F(B)$ for any $A \subseteq B \subseteq V$. However, many important applications in machine learning lead to *non-monotone* submodular functions. For example, graphical model inference [5, 17], or trading off any submodular gain maximization with costs (functions of the form $F(S) = G(S) - \lambda M(S)$, where $G(S)$ is monotone submodular and $M(S)$ a linear (modular) cost function), such as for utility-privacy tradeoffs [18], require maximizing non-monotone submodular functions. For non-monotone functions, the simple greedy algorithm in [8] can perform arbitrarily poorly (see Appendix H.1 for an example). Intuitively, the introduction of additional elements with monotone submodular functions never decreases the objective while introducing elements with non-monotone submodular functions can *decrease* the objective to its minimum. For non-monotone functions, Buchbinder et al. [2] recently proposed an optimal double greedy algorithm that works well in a serial setting. In this paper, we study parallelizations of this algorithm.

**The serial double greedy algorithm.**   The serial double greedy algorithm of Buchbinder et al. [2] (Ser-2g, in Alg. 3) maintains two sets $A^i \subseteq B^i$. Initially, $A^0 = \emptyset$ and $B^0 = V$. In iteration $i$, the set $A^{i-1}$ contains the items selected before item/iteration $i$, and $B^{i-1}$ contains $A^i$ and the items that are so far undecided. The algorithm serially passes through the items in $V$ and determines online whether to keep item $i$ (add to $A^i$) or discard it (remove from $B^i$), based on a threshold that trades off the gain $\Delta_+(i) = F(A^{i-1} \cup i) - F(A^{i-1})$ of adding $i$ to the currently selected set $A^{i-1}$, and the gain $\Delta_-(i) = F(B^{i-1} \setminus i) - F(B^{i-1})$ of removing $i$ from the candidate set, estimating its complementarity to other remaining elements. For any element ordering, this algorithm achieves a tight 1/2-approximation in expectation.

## 3   Concurrency Patterns for Parallel Machine Learning

In this paper we adopt a transactional view of the program state and explore parallelization strategies through the lens of parallel transaction processing systems. We recast the program state (the sets $A$ and $B$) as data, and the operations (adding elements to $A$ and removing elements from $B$) as

transactions. More precisely we reformulate the double greedy algorithm (Alg. 3) as a series of *exchangeable*, *Read-Write* transactions of the form:

$$T_e(A, B) \triangleq \begin{cases} (A \cup e, B) & \text{if } u_e \leq \frac{[\Delta_+(A,e)]_+}{[\Delta_+(A,e)]_+ + [\Delta_-(B,e)]_+} \\ (A, B \backslash e) & \text{otherwise.} \end{cases} \tag{1}$$

The transaction $T_e$ is a function from the sets $A$ and $B$ to new sets $A$ and $B$ based on the element $e \in V$ and the predetermined random bits $u_e$ for that element.

By composing the transactions $T_n(T_{n-1}(\dots T_1(\emptyset, V)))$ we recover the serial double-greedy algorithm defined in Alg. 3. In fact, any ordering of the *serial* composition of the transactions recovers a permuted execution of Alg. 3 and therefore the optimal approximation algorithm. However, this raises the question: *is it possible to apply transactions in parallel?* If we execute transactions $T_i$ and $T_j$, with $i \neq j$, in parallel we need a method to merge the resulting program states. In the context of the double greedy algorithm, we could define the parallel execution of two transactions as:

$$T_i(A, B) + T_j(A, B) \triangleq (T_i(A, B)_A \cup T_j(A, B)_A, \; T_i(A, B)_B \cap T_j(A, B)_B), \tag{2}$$

the union of the resulting $A$ and the intersection of the resulting $B$. While we can easily generalize Eq. (2) to many parallel transactions, we cannot always guarantee that the result will correspond to a serial composition of transactions. As a consequence, we cannot directly apply the analysis of Buchbinder et al. [2] to derive strong approximation guarantees for the parallel execution.

Fortunately, several decades of research [19, 20] in database systems have explored efficient parallel transaction processing. In this paper we adopt a coordinated bounds approach to parallel transaction processing in which parallel transactions are constructed under bounds on the possible program state. If the transaction could violate the bound then it is processed serially on the server. By adjusting the definition of the bound we can span a space of coordination-free to serializable executions.

| **Algorithm 1:** Generalized transactions | **Algorithm 2:** Commit transaction $i$ |
|---|---|
| 1 **for** $p \in \{1, \dots, P\}$ **do in parallel** | 1 **wait until** $\forall j < i$, processed$(j) = true$ |
| 2     **while** $\exists$ *element to process* **do** | 2 **Atomically** |
| 3        $e = $ next element to process | 3     **if** $\partial_i = $ *FAIL* **then** |
| 4        $(\mathfrak{g}_e, i) = $ requestGuarantee$(e)$ |        // Deferred proposal |
| 5        $\partial_i = $ propose$(e, \mathfrak{g}_e)$ | 4        $\partial_i = $ propose$(e, \mathfrak{S})$ |
| 6        commit$(e, i, \partial_i)$ // Non-blocking |     // Advance the program state |
| | 5     $\mathfrak{S} \leftarrow \partial_i(\mathfrak{S})$ |

Figure 1: Algorithm for generalized transactions. Each transaction requests its position $i$ in the commit ordering, as well as the bounds $\mathfrak{g}_e$ that are guaranteed to hold when it commits. Transactions are also guaranteed to be committed according to the given ordering.

In Fig. 1 we describe the coordinated bounds transaction pattern. The clients (Alg. 1), in parallel, construct and commit transactions under bounded assumptions about the program state $\mathfrak{S}$ (*i.e.,* the sets $A$ and $B$). Transactions are constructed by requesting the latest bound $\mathfrak{g}_e$ on $\mathfrak{S}$ at logical time $i$ and computing a change $\partial_i$ to $\mathfrak{S}$ (*e.g.,* Add $e$ to A). If the bound is insufficient to construct the transaction then $\partial_i = $ FAIL is returned. The client then sends the proposed change $\partial_i$ to the server to be committed atomically and proceeds to the next element without waiting for a response.

The server (Alg. 2) *serially* applies the transactions advancing the program state (*i.e.,* adding elements to $A$ or removing elements from $B$). If the bounds were insufficient and the transaction failed at the client (*i.e.,* $\partial_i = $ FAIL) then the server *serially* reconstructs and applies the transaction under the true program state. Moreover, the server is responsible for deriving bounds, processing transactions in the logical order $i$, and producing the serializable output $\partial_n(\partial_{n-1}(\dots \partial_1(\mathfrak{S})))$.

This model achieves a high degree of parallelism when the cost of constructing the transaction dominates the cost of applying the transaction. For example, in the case of submodular maximization, the cost of constructing the transaction depends on evaluating the marginal gains with respect to changes in $A$ and $B$ while the cost of applying the transaction reduces to setting a bit. It is also essential that only a few transactions fail at the client. Indeed, the analysis of these systems focuses on ensuring that the majority of the transactions succeed.

---

**Algorithm 3:** Ser-2g: serial double greedy

1   $A^0 = \emptyset, B^0 = V$
2   **for** $i = 1$ *to* $n$ **do**
3     $\Delta_+(i) = F(A^{i-1} \cup i) - F(A^{i-1})$
4     $\Delta_-(i) = F(B^{i-1} \backslash i) - F(B^{i-1})$
5     Draw $u_i \sim Unif(0, 1)$
6     **if** $u_i < \frac{[\Delta_+(i)]_+}{[\Delta_+(i)]_+ + [\Delta_-(i)]_+}$ **then**
7       $A^i := A^{i-1} \cup i; B^i := B^{i-1}$
8     **else** $A^i := A^{i-1}; B^i := B^{i-1} \backslash i$

---

**Algorithm 4:** CF-2g: coord-free double greedy

1   $\widehat{A} = \emptyset, \widehat{B} = V$
2   **for** $p \in \{1, \ldots, P\}$ **do in parallel**
3     **while** $\exists$ *element to process* **do**
4       $e =$ next element to process
5       $\widehat{A}_e = \widehat{A}; \widehat{B}_e = \widehat{B}$
6       $\Delta_+^{\max}(e) = F(\widehat{A}_e \cup e) - F(\widehat{A}_e)$
7       $\Delta_-^{\max}(e) = F(\widehat{B}_e \backslash e) - F(\widehat{B}_e)$
8       Draw $u_e \sim Unif(0, 1)$
9       **if** $u_e < \frac{[\Delta_+^{\max}(e)]_+}{[\Delta_+^{\max}(e)]_+ + [\Delta_-^{\max}(e)]_+}$ **then**
10        $\widehat{A}(e) \leftarrow 1$
11       **else** $\widehat{B}(e) \leftarrow 0$

---

**Algorithm 5:** CC-2g: concurrency control

1   $\widehat{A} = \widetilde{A} = \emptyset, \widehat{B} = \widetilde{B} = V$
2   **for** $i = 1, \ldots, |V|$ **do** processed$(i) = false$
3   $\iota = 0$
4   **for** $p \in \{1, \ldots, P\}$ **do in parallel**
5     **while** $\exists$ *element to process* **do**
6       $e =$ next element to process
7       $(\widehat{A}_e, \widetilde{A}_e, \widehat{B}_e, \widetilde{B}_e, i) =$ getGuarantee$(e)$
8       (result, $u_e$) = propose$(e, \widehat{A}_e, \widetilde{A}_e, \widehat{B}_e, \widetilde{B}_e)$
9       commit$(e, i, u_e,$ result$)$

---

**Algorithm 6:** CC-2g getGuarantee$(e)$

1   $\widetilde{A}(e) \leftarrow 1; \widetilde{B}(e) \leftarrow 0$
2   $i = \iota; \iota \leftarrow \iota + 1$
3   $\widehat{A}_e = \widehat{A}; \widehat{B}_e = \widehat{B}$
4   $\widetilde{A}_e = \widetilde{A}; \widetilde{B}_e = \widetilde{B}$
5   **return** $(\widehat{A}_e, \widetilde{A}_e, \widehat{B}_e, \widetilde{B}_e, i)$

---

**Algorithm 7:** CC-2g propose

1   $\Delta_+^{\min}(e) = F(\widetilde{A}_e) - F(\widetilde{A}_e \backslash e)$
2   $\Delta_+^{\max}(e) = F(\widehat{A}_e \cup e) - F(\widehat{A}_e)$
3   $\Delta_-^{\min}(e) = F(\widetilde{B}_e) - F(\widetilde{B}_e \cup e)$
4   $\Delta_-^{\max}(e) = F(\widehat{B}_e \backslash e) - F(\widehat{B}_e)$
5   Draw $u_e \sim Unif(0, 1)$
6   **if** $u_e < \frac{[\Delta_+^{\min}(e)]_+}{[\Delta_+^{\min}(e)]_+ + [\Delta_-^{\max}(e)]_+}$ **then**
7     result $\leftarrow 1$
8   **else if** $u_e > \frac{[\Delta_+^{\max}(e)]_+}{[\Delta_+^{\max}(e)]_+ + [\Delta_-^{\min}(e)]_+}$ **then**
9     result $\leftarrow -1$
10   **else** result $\leftarrow$ FAIL
11   **return** (result, $u_e$)

---

**Algorithm 8:** CC-2g: commit$(e, i, u_e,$ result$)$

1   **wait until** $\forall j < i$, processed$(j) = true$
2   **if** *result = FAIL* **then**
3     $\Delta_+^{exact}(e) = F(\widehat{A} \cup e) - F(\widehat{A})$
4     $\Delta_-^{exact}(e) = F(\widehat{B} \backslash e) - F(\widehat{B})$
5     **if** $u_e < \frac{[\Delta_+^{exact}(e)]_+}{[\Delta_+^{exact}(e)]_+ + [\Delta_-^{exact}(e)]_+}$ **then** result $\leftarrow 1$
6     **else** result $\leftarrow -1$
7   **if** *result = 1* **then** $\widehat{A}(e) \leftarrow 1; \widetilde{B}(e) \leftarrow 1$
8   **else** $\widetilde{A}(e) \leftarrow 0; \widehat{B}(e) \leftarrow 0$
9   processed$(i) = true$

---

## 4   Coordination-Free Double Greedy Algorithm

The coordination-free approach attempts to reduce the need to coordinate guarantees and the logical ordering. This is achieved by operating on potentially stale states: the transaction guarantee reduces to requiring $\mathfrak{g}_e$ be a stale version of $\mathfrak{S}$, and the logical ordering is implicitly defined by the time of commit. In using these weak guarantees, CF-2g is overly optimistically assuming that concurrent transactions are independent, which could potentially lead to erroneous decisions.

Alg. 4 is the coordination-free parallel double greedy algorithm.[1] CF-2g closely resembles the serial Ser-2g, but the elements $e \in V$ are no longer processed in a fixed order. Thus, the sets $A, B$ are replaced by potentially stale local estimates (bounds) $\widehat{A}, \widehat{B}$, where $\widehat{A}$ is a subset of the true $A$ and $\widehat{B}$ is a superset of the actual $B$ on each iteration. These bounding sets allow us to compute bounds $\Delta_+^{\max}, \Delta_-^{\max}$ which approximate $\Delta_+, \Delta_-$ from the serial algorithm. We now formalize this idea.

To analyze the CF-2g algorithm we order the elements $e \in V$ according to the commit time (*i.e.,* when Alg. 4 line 8 is executed). Let $\iota(e)$ be the position of $e$ in this total ordering on elements. This

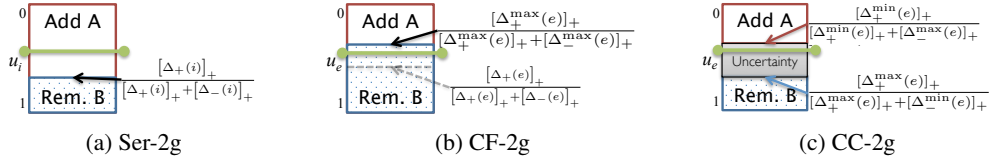

(a) Ser-2g            (b) CF-2g            (c) CC-2g

Figure 2: Illustration of algorithms. **(a)** Ser-2g computes a threshold based on the true values $\Delta_+$, $\Delta_-$, and chooses an action based by comparing a uniform random $u_i$ against the threshold. **(b)** CF-2g approximates the threshold based on stale $\widehat{A}$, $\widehat{B}$, possibly choosing the wrong action. **(c)** CC-2g computes two thresholds based on the bounds on $A$, $B$, which defines an uncertainty region where it is not possible to choose the correct action locally. If the random value $u_e$ falls inside the uncertainty interval than the transaction FAILS and must be recomputed serially by the server; otherwise the transaction holds under all possible global states.

ordering allows us to define monotonically non-decreasing sets $A^i = \{e' : e' \in A, \iota(e') < i\}$ where $A$ is the final returned set, and monotonically non-increasing sets $B^i = A^i \cup \{e' : \iota(e') \geq i\}$. The sets $A^i, B^i$ provide a serialization against which we can compare CF-2g; in this serialization, Alg. 3 computes $\Delta_+(e) = F(A^{\iota(e)-1} \cup e) - F(A^{\iota(e)-1})$ and $\Delta_-(e) = F(B^{\iota(e)-1}\backslash e) - F(B^{\iota(e)-1})$. On the other hand, CF-2g uses stale versions[2] $\widehat{A}_e$, $\widehat{B}_e$: Alg. 4 computes $\Delta_+^{\max}(e) = F(\widehat{A}_e \cup e) - F(\widehat{A}_e)$ and $\Delta_-^{\max}(e) = F(\widehat{B}_e\backslash e) - F(\widehat{B}_e)$.

The next lemma shows that $\widehat{A}_e$, $\widehat{B}_e$ are bounding sets for the serialization's sets $A^{\iota(e)-1}, B^{\iota(e)-1}$. Intuitively, the bounds hold because $\widehat{A}_e$, $\widehat{B}_e$ are stale versions of $A^{\iota(e)-1}$, $B^{\iota(e)-1}$, which are monotonically non-decreasing and non-increasing sets. Appendix A gives a detailed proof.

**Lemma 4.1.** *In CF-2g, for any $e \in V$, $\widehat{A}_e \subseteq A^{\iota(e)-1}$, and $\widehat{B}_e \supseteq B^{\iota(e)-1}$.*

**Corollary 4.2.** *Submodularity of $F$ implies for CF-2g $\Delta_+(e) \leq \Delta_+^{\max}(e)$, and $\Delta_-(e) \leq \Delta_-^{\max}(e)$.*

The error in CF-2g depends on the tightness of the bounds in Cor. 4.2. We analyze this in Sec. 6.1.

## 5    Concurrency Control for the Double Greedy Algorithm

The concurrency control-based double greedy algorithm[1], CC-2g, is presented in Alg. 5, and closely follows the meta-algorithm of Alg. 1 and Alg. 2. Unlike in CF-2g, the concurrency control mechanisms of CC-2g ensure that concurrent transactions are serialized when they are not independent.

Serializability is achieved by maintaining sets $\widehat{A}$, $\widetilde{A}$, $\widehat{B}$, $\widetilde{B}$, which serve as upper *and* lower bounds on the true state of $A$ and $B$ at commit time. Each thread can determine locally if a decision to include or exclude an element can be taken safely. Otherwise, the proposal is deferred to the commit process (Alg. 8) which waits until it is certain about $A$ and $B$ before proceeding.

The commit order is given by $\iota(e)$, which is the value of $\iota$ in line 2 of Alg. 5. We define $A^{\iota(e)-1}$, $B^{\iota(e)-1}$ as before with CF-2g. Additionally, let $\widehat{A}_e$, $\widehat{B}_e$, $\widetilde{A}_e$, and $\widetilde{B}_e$ be the sets that are returned by Alg. 6.[2] Indeed, these sets are guaranteed to be bounds on $A^{\iota(e)-1}, B^{\iota(e)-1}$:

**Lemma 5.1.** *In CC-2g, $\forall e \in V$, $\widehat{A}_e \subseteq A^{\iota(e)-1} \subseteq \widetilde{A}_e\backslash e$, and $\widehat{B}_e \supseteq B^{\iota(e)-1} \supseteq \widetilde{B}_e \cup e$.*

Intuitively, these bounds are maintained by recording potential effects of concurrent transactions in $\widetilde{A}$, $\widetilde{B}$, and only recording the actual effects in $\widehat{A}$, $\widehat{B}$; we leave the full proof to Appendix A. Furthermore, by committing transactions in order $\iota$, we have $\widehat{A} = A^{\iota(e)-1}$ and $\widehat{B} = B^{\iota(e)-1}$ during commit.

**Lemma 5.2.** *In CC-2g, when committing element $e$, we have $\widehat{A} = A^{\iota(e)-1}$ and $\widehat{B} = B^{\iota(e)-1}$.*

**Corollary 5.3.** *Submodularity of $F$ implies that the $\Delta$'s computed by CC-2g satisfy $\Delta_+^{\min}(e) \leq \Delta_+^{exact}(e) = \Delta_+(e) \leq \Delta_+^{\max}(e)$ and $\Delta_-^{\min}(e) \leq \Delta_-^{exact}(e) = \Delta_-(e) \leq \Delta_-^{\max}(e)$.*

By using these bounds, CC-2g can determine when it is safe to construct the transaction locally. For failed transactions, the server is able to construct the correct transaction using the true program state. As a consequence we can guarantee that the parallel execution of CC-2g is serializable.

# 6 Analysis of Algorithms

Our two algorithms trade off performance and strong approximation guarantees. The CF-2g algorithm emphasizes speed at the expense of the approximation objective. On the other hand, CC-2g emphasizes the tight $1/2$-approximation at the expense of increased coordination. In this section we characterize the reduction in the approximation objective as well as the increased coordination. Our analysis connects the degradation in CC-2g scalability with the degradation in the CF-2g approximation factor via the maximum inter-processor message delay $\tau$.

## 6.1 Approximation of CF-2g double greedy

**Theorem 6.1.** *Let $F$ be a non-negative submodular function. CF-2g solves the unconstrained problem $\max_{A \subset V} F(A)$ with worst-case approximation factor $E[F(A_{CF})] \geq \frac{1}{2}F^* - \frac{1}{4}\sum_{i=1}^N E[\rho_i]$, where $A_{CF}$ is the output of the algorithm, $F^*$ is the optimal value, and $\rho_i = \max\{\Delta_+^{\max}(e) - \Delta_+(e), \Delta_-^{\max}(e) - \Delta_-(e)\}$ is the maximum discrepancy in the marginal gain due to the bounds.*

The proof (Appendix C) of Thm. 6.1 follows the structure in [2]. Thm. 6.1 captures the deviation from optimality as a function of width of the bounds which we characterize for two common applications.

**Example: max graph cut.** For the max cut objective we bound the expected discrepancy in the marginal gain $\rho_i$ in terms of the sparsity of the graph and the maximum inter-processor message delay $\tau$. By applying Thm. 6.1 we obtain the approximation factor $E[F(A^N)] \geq \frac{1}{2}F^* - \tau\frac{\#\text{edges}}{2N}$ which decreases linearly in both the message delays and graph density. In a complete graph, $F^* = \frac{1}{2}\#\text{edges}$, so $E[F(A^N)] \geq F^*\left(\frac{1}{2} - \frac{\tau}{N}\right)$, which makes it possible to scale $\tau$ linearly with $N$ while retaining the same approximation factor.

**Example: set cover.** Consider the simple set cover function, $F(A) = \sum_{l=1}^L \min(1, |A \cap S_l|) - \lambda|A| = |\{l : A \cap S_l \neq \emptyset\}| - \lambda|A|$, with $0 < \lambda \leq 1$. We assume that there is some bounded delay $\tau$. Suppose also the $S_l$'s form a partition, so each element $e$ belongs to exactly one set. Then, $\sum_e E[\rho_e] \geq \tau + L(1 - \lambda^\tau)$, which is linear in $\tau$ but independent of $N$.

## 6.2 Correctness of CC-2g

**Theorem 6.2.** *CC-2g is serializable and therefore solves the unconstrained submodular maximization problem $\max_{A \subset V} F(A)$ with approximation $E[F(A_{CC})] \geq \frac{1}{2}F^*$, where $A_{CC}$ is the output of the algorithm, and $F^*$ is the optimal value.*

The key challenge in the proof (Appendix B) of Thm. 6.2 is to demonstrate that CC-2g guarantees a serializable execution. It suffices to show that CC-2g takes the same decision as Ser-2g for each element – locally if it is safe to do so, and otherwise deferring the computation to the server. As an immediate consequence of serializability, we recover the optimal approximation guarantees of the serial Ser-2g algorithm.

## 6.3 Scalability of CC-2g

Whenever a transaction is reconstructed on the server, the server needs to wait for all earlier elements to be committed, and is also blocked from committing all later elements. Each failed transaction effectively constitutes a barrier to the parallel processing. Hence, the scalability of CC-2g is dependent on the number of failed transactions.

We can directly bound the number of failed transactions (details in Appendix D) for both the max-cut and set cover example problems. For the max-cut problem with a maximum inter-processor message

delay $\tau$ we obtain the upper bound $2\tau\frac{\#edges}{N}$. Similarly for set cover the expected *number* of failed transactions is upper-bounded by $2\tau$. As a consequence, the coordination costs of CC-2g grows at the same rate as the reduction in accuracy of CF-2g. Moreover, the CC-2g algorithm will slow down in settings where the CF-2g algorithm produces sub-optimal solutions.

## 7    Evaluation

We implemented the parallel and serial double greedy algorithms in Java / Scala. Experiments were conducted on Amazon EC2 using one cc2.8xlarge machine, up to 16 threads, for 10 repetitions. We measured the runtime and speedup (ratio of runtime on 1 thread to runtime on $p$ threads). For CF-2g, we measured $F(A_{CF}) - F(A_{Ser})$, the difference between the objective value on the sets returned by CF-2g and Ser-2g. We verified the correctness of CC-2g by comparing the output of CC-2g with Ser-2g. We also measured the fraction of transactions that fail in CC-2g. Our parallel algorithms were tested on the max graph cut and set cover problems with two synthetic graphs and three real datasets (Table 1). We found that vertices were typically indexed such that nearby vertices in the graph were also close in their indices. To reduce this dependency, we randomly permuted the ordering of vertices.

| Graph | # vertices | # edges | Description |
|---|---|---|---|
| Erdos-Renyi | 20,000,000 | $\approx 2 \times 10^9$ | Each edge is included with probability $5 \times 10^{-6}$. |
| ZigZag | 25,000,000 | 2,025,000,000 | Expander graph. The 81-regular zig-zag product between the Cayley graph on $\mathbb{Z}_{2500000}$ with generating set $\{\pm 1, \ldots, \pm 5\}$, and the complete graph $K_{10}$. |
| Friendster | 10,000,000 | 625,279,786 | Subgraph of social network. [21] |
| Arabic-2005 | 22,744,080 | 631,153,669 | 2005 crawl of Arabic web sites [22, 23, 24]. |
| UK-2005 | 39,459,925 | 921,345,078 | 2005 crawl of the .uk domain [22, 23, 24]. |
| IT-2004 | 41,291,594 | 1,135,718,909 | 2004 crawl of the .it domain [22, 23, 24]. |

Table 1: Synthetic and real graphs used in the evaluation of our parallel algorithms.

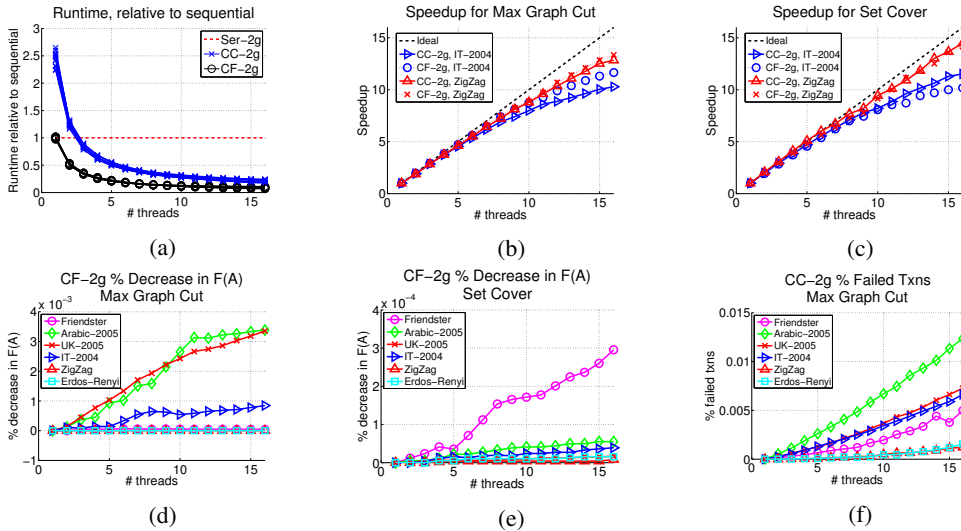

Figure 3: Experimental results. Fig. 3a – runtime of the parallel algorithms as a ratio to that of the serial algorithm. Each curve shows the runtime of a parallel algorithm on a particular graph for a particular function $F$. Fig. 3b, 3c – speedup (ratio of runtime on one thread to that on $p$ threads). Fig. 3d, 3e – % difference between objective values of Ser-2g and CF-2g, i.e. $[F(A_{CF})/F(A_{Ser}) - 1] \times 100\%$. Fig. 3f – percentage of transactions that fail in CC-2g on the max graph cut problem.

We summarize of the key results here with more detailed experiments and discussion in Appendix G. **Runtime, Speedup:** Both parallel algorithms are faster than the serial algorithm with three or more threads, and show good speedup properties as more threads are added ($\sim 10\text{x}$ or more for all graphs and both functions). **Objective value:** The objective value of CF-2g decreases with the number of threads, but differs from the serial objective value by less than $0.01\%$. **Failed transactions:** CC-2g fails more transactions as threads are added, but even with 16 threads, less than $0.015\%$ transactions fail, which has negligible effect on the runtime / speedup.

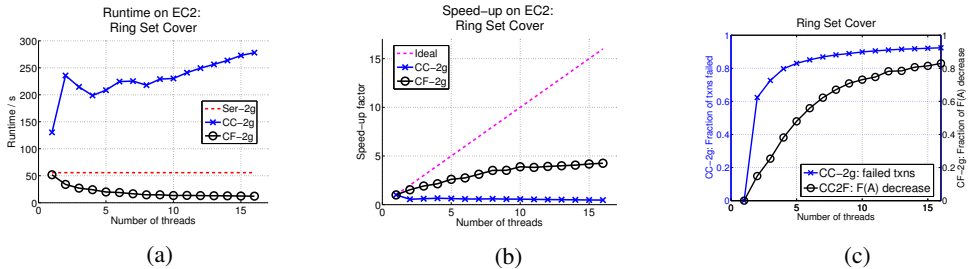

Figure 4: Experimental results for set cover problem on a ring expander graph demonstrating that for adversarially constructed inputs we can reduce the optimality of CF-2g and increase coordination costs for CC-2g.

## 7.1 Adversarial ordering

To highlight the differences in approaches between the two parallel algorithms, we conducted experiments on a ring Cayley expander graph on $\mathbb{Z}_{10^6}$ with generating set $\{\pm 1, \ldots, \pm 1000\}$. The algorithms are presented with an adversarial ordering, without permutation, so vertices close in the ordering are adjacent to one another, and tend to be processed concurrently. This causes CF-2g to make more mistakes, and CC-2g to fail more transactions. While more sophisticated partitioning schemes could improve scalability and eliminate the effect of adversarial ordering, we use the default data partitioning in our experiments to highlight the differences between the two algorithms. As Fig. 4 shows, CC-2g sacrifices speed to ensure a serializable execution, eventually failing on $> 90\%$ of transactions. On the other hand, CF-2g focuses on speed, resulting in faster runtime, but achieves an objective value that is $20\%$ of $F(A_{Ser})$. We emphasize that we contrived this example to highlight differences between CC-2g and CF-2g, and we do not expect to see such orderings in practice.

## 8 Related Work

**Similar approach:** Coordination-free solutions have been proposed for stochastic gradient descent [25] and collapsed Gibbs sampling [26]. More generally, parameter servers [27, 28] apply the CF approach to larger classes of problems. Pan et al. [29] applied concurrency control to parallelize some unsupervised learning algorithms. **Similar problem:** Distributed and parallel greedy submodular maximization is addressed in [1, 15, 16], but only for monotone functions.

## 9 Conclusion and Future Work

By adopting the transaction processing model from parallel database systems, we presented two approaches to parallelizing the double greedy algorithm for unconstrained submodular maximization. We quantified the weaker approximation guarantee of CF-2g and the additional coordination of CC-2g, allowing one to trade off between performance and objective optimality. Our evaluation on large scale data demonstrates the scalability and tradeoffs of the two approaches. Moreover, as the approximation quality of the CF-2g algorithm decreases so does the scalability of the CC-2g algorithm. The choice between the algorithm then reduces to a choice of guaranteed performance and guaranteed optimality.

We believe there are a number of areas for future work. One can imagine a system that allows a smooth interpolation between CF-2g and CC-2g. While both CF-2g and CC-2g can be immediately implemented as distributed algorithms, higher communication costs and delays may pose additional challenges. Finally, other problems such as constrained maximization of monotone / non-monotone functions could potentially be parallelized with the CF and CC frameworks.

**Acknowledgments.** This research is supported in part by NSF CISE Expeditions Award CCF-1139158, LBNL Award 7076018, and DARPA XData Award FA8750-12-2-0331, and gifts from Amazon Web Services, Google, SAP, The Thomas and Stacey Siebel Foundation, Adobe, Apple, Inc., Bosch, C3Energy, Cisco, Cloudera, EMC, Ericsson, Facebook, GameOnTalis, Guavus, HP, Huawei, Intel, Microsoft, NetApp, Pivotal, Splunk, Virdata, VMware, and Yahoo!. This research was in part funded by the Office of Naval Research under contract/grant number N00014-11-1-0688. X. Pan's work is also supported by a DSO National Laboratories Postgraduate Scholarship.

## Footnotes

[1]We present only the parallelized probabilistic versions of [2]. Both parallel algorithms can be easily extended to the deterministic version of [2]; CF-2g can also be extended to the multilinear version of [2].

[2] For clarity, we present the algorithm as creating a copy of $\widehat{A}$, $\widehat{B}$, $\widetilde{A}$, and $\widetilde{B}$ for each element. In practice, it is more efficient to update and access them in shared memory. Nevertheless, our theorems hold for both settings.

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
