[Supplementary Material]

# A  Proofs of $\widetilde{A}_e$, $\widehat{A}_e$, $\widetilde{B}_e$, $\widehat{B}_e$ as bounds on $A^{\iota(e)-1}$ and $B^{\iota(e)-1}$

**Lemma 4.1.** *In CF-2g, for any $e \in V$, $\widehat{A}_e \subseteq A^{\iota(e)-1}$, and $\widehat{B}_e \supseteq B^{\iota(e)-1}$.*

*Proof.* For any element $e$, we write $T_e$ to denote the time at which Alg. 4 line 8 is executed. Consider any element $e' \in V$. If $e' \in \widehat{A}_e$, it must be the case that the algorithm set $\widehat{A}(e')$ to 1 (line 10) before $T_e$, which implies $\iota(e') < \iota(e)$, and hence $e' \in A^{\iota(e)-1}$. So $\widehat{A}_e \subseteq A^{\iota(e)-1}$.

Similarly, if $e' \notin \widehat{B}_e$, then the algorithm set $\widehat{B}(e')$ to 0 (line 11) before $T_e$, so $\iota(e') < \iota(e)$. Also, $e' \notin A$ because the execution of line 11 excludes the execution of line 10. Therefore, $e' \notin A^{\iota(e)-1}$, and $e' \notin B^{\iota(e)-1}$. So $\widehat{B}_e \supseteq B^{\iota(e)-1}$. □

**Lemma 5.1.** *In CC-2g, $\forall e \in V$, $\widehat{A}_e \subseteq A^{\iota(e)-1} \subseteq \widetilde{A}_e \backslash e$, and $\widehat{B}_e \supseteq B^{\iota(e)-1} \supseteq \widetilde{B}_e \cup e$.*

*Proof.* Clearly, $e \in \widetilde{B}_e \cup e$ but $e \notin \widetilde{A}_e \backslash e$. By definition, $e \in B^{\iota(e)-1}$ but $e \notin A^{\iota(e)-1}$. CC-2g only modifies $\widehat{A}(e)$ and $\widehat{B}(e)$ when committing the transaction on $e$, which occurs after obtaining the bounds in getGuarantee($e$), so $e \in \widehat{B}_e$ but $e \notin \widehat{A}_e$.

Consider any $e' \neq e$. Suppose $e' \in \widehat{A}_e$. This is only possible if we have committed the transaction on $e'$ before the call getGuarantee($e$), so it must be the case that $\iota(e') < \iota(e)$. Thus, $e' \in A^{\iota(e)-1}$.

Now suppose $e' \in A^{\iota(e)-1}$. By definition, this implies $\iota(e') < \iota(e)$ and $e' \in A$. Hence, it must be the case that we have already set $\widetilde{A}(e') \leftarrow 1$ (by the ordering imposed by $\iota$ on Line 2), but never execute $\widetilde{A}(e') \leftarrow 0$ (since $e' \in A$), so $e' \in \widetilde{A}_e$.

An analogous argument shows $e' \notin \widehat{B}_e \implies e' \notin B^{\iota(e)-1} \implies e' \notin \widetilde{B}_e \cup e$. □

**Lemma 5.2.** *In CC-2g, when committing element $e$, we have $\widehat{A} = A^{\iota(e)-1}$ and $\widehat{B} = B^{\iota(e)-1}$.*

*Proof.* Alg. 8 Line 1 ensures that all elements ordered before $e$ are committed, and that no element ordered after $e$ are committed. This suffices to guarantee that $e' \in \widehat{A} \iff e' \in A^{\iota(e)-1}$ and $e' \in \widehat{B} \iff e' \in B^{\iota(e)-1}$. □

# B  Proof of serial equivalence of CC-2g

**Theorem 6.2.** *CC-2g is serializable and therefore solves the unconstrained submodular maximization problem $\max_{A \subset V} F(A)$ with approximation $E[F(A_{CC})] \geq \frac{1}{2}F^*$, where $A_{CC}$ is the output of the algorithm, and $F^*$ is the optimal value.*

*Proof.* We will denote by $A_{seq}^i$, $B_{seq}^i$ the sets generated by Ser-2g, reserving $A^i$, $B^i$ for sets generated by the CC-2g algorithm. It suffices to show by induction that $A_{seq}^i = A^i$ and $B_{seq}^i = B^i$. For the base case, $A^0 = \emptyset = A_{seq}^0$, and $B^0 = V = B_{seq}^0$. Consider any element $e$. The CC-2g algorithm includes $e \in A$ iff $u_e < [\Delta_+^{\min}(e)]_+([\Delta_+^{\min}(e)]_+ + [\Delta_-^{\max}(e)]_+)^{-1}$ on Alg. 5 Line 6 or $u_e < [\Delta_+^{\text{exact}}(e)]_+([\Delta_+^{\text{exact}}(e)]_+ + [\Delta_-^{\text{exact}}(e)]_+)^{-1}$ on Alg. 8 Line 5. In both cases, Corollary 5.3 implies $u_e < [\Delta_+(e)]_+([\Delta_+(e)]_+ + [\Delta_-(e)]_+)^{-1}$. By induction, $A^{\iota(e)-1} = A_{seq}^{\iota(e)-1}$ and $B^{\iota(e)-1} = B_{seq}^{\iota(e)-1}$, so the threshold is exactly that computed by Ser-2g. Hence, the CC-2g algorithm includes $e \in A$ iff Ser-2g includes $e \in A$. (An analogous argument works for the case where $e$ is excluded from $B$.) □

# C Proof of bound for CF-2g

We follow the proof outline of [2].

Consider an ordering $\iota$ inducted by running CF-2g. For convenience, we will use $i$ to flexibly denote the element $e$ and its ordering $\iota(e)$.

Let $OPT$ be an optimal solution to the problem. Define $O^i := (OPT \cup A^i) \cap B^i$. Note that $O^i$ coincides with $A^i$ and $B^i$ on elements $1, \ldots, i$, and $O^i$ coincides with $OPT$ on elements $i+1, \ldots, n$. Hence,

$$O^i \backslash (i+1) \supseteq A^i$$
$$O^i \cup (i+1) \subseteq B^i.$$

**Lemma C.1.** *For every* $1 \le i \le n$, $\Delta_+(i) + \Delta_-(i) \ge 0$.

*Proof.* This is just Lemma II.1 of [2]. $\qquad\square$

**Lemma C.2.** *Let* $\rho_i = \max\{\Delta_+^{\max}(e) - \Delta_+(e), \Delta_-^{\max}(e) - \Delta_-(e)\}$. *For every* $1 \le i \le n$,

$$E[F(O^{i-1}) - F(O^i)] \le \frac{1}{2} E[F(A^i) - F(A^{i-1}) + F(B^i) - F(B^{i-1}) + \rho_i].$$

*Proof.* We follow the proof outline of [2]. First, note that it suffices to prove the inequality conditioned on knowing $A^{i-1}$, $\widehat{A}_i$ and $\widehat{B}_i$, then applying the law of total expectation. Under this conditioning, we also know $B^{i-1}$, $O^{i-1}$, $\Delta_+(i)$, $\Delta_+^{\max}(i)$, $\Delta_-(i)$, and $\Delta_-^{\max}(i)$.

We consider the following 6 cases.

**Case 1:** $0 < \Delta_+(i) \le \Delta_+^{\max}(i)$, $0 \le \Delta_-^{\max}(i)$. Since both $\Delta_+^{\max}(i) > 0$ and $\Delta_-^{\max}(i) > 0$, the probability of including $i$ is just $\Delta_+^{\max}(i)/(\Delta_+^{\max}(i) + \Delta_-^{\max}(i))$, and the probability of excluding $i$ is $\Delta_-^{\max}(i)/(\Delta_+^{\max}(i) + \Delta_-^{\max}(i))$.

$$
\begin{aligned}
E[F(A^i) - F(A^{i-1})|A^{i-1}, \widehat{A}_i, \widehat{B}_i] &= \frac{\Delta_+^{\max}(i)}{\Delta_+^{\max}(i) + \Delta_-^{\max}(i)}(F(A^{i-1} \cup i) - F(A^{i-1})) \\
&= \frac{\Delta_+^{\max}(i)}{\Delta_+^{\max}(i) + \Delta_-^{\max}(i)}\Delta_+(i) \\
&\ge \frac{\Delta_+^{\max}(i)}{\Delta_+^{\max}(i) + \Delta_-^{\max}(i)}(\Delta_+^{\max}(i) - \rho_i) \\
E[F(B^i) - F(B^{i-1})|A^{i-1}, \widehat{A}_i, \widehat{B}_i] &= \frac{\Delta_-^{\max}(i)}{\Delta_+^{\max}(i) + \Delta_-^{\max}(i)}(F(B^{i-1} \backslash i) - F(B^{i-1})) \\
&= \frac{\Delta_-^{\max}(i)}{\Delta_+^{\max}(i) + \Delta_-^{\max}(i)}\Delta_-(i) \\
&\ge \frac{\Delta_-^{\max}(i)}{\Delta_+^{\max}(i) + \Delta_-^{\max}(i)}(\Delta_-^{\max}(i) - \rho_i)
\end{aligned}
$$

$$E[F(O^{i-1}) - F(O^i)|A^{i-1}, \widehat{A}_i, \widehat{B}_i]$$

$$= \frac{\Delta_+^{\max}(i)}{\Delta_+^{\max}(i) + \Delta_-^{\max}(i)}(F(O^{i-1}) - F(O^{i-1} \cup i))$$

$$+ \frac{\Delta_-^{\max}(i)}{\Delta_+^{\max}(i) + \Delta_-^{\max}(i)}(F(O^{i-1}) - F(O^{i-1}\backslash i))$$

$$= \begin{cases} \frac{\Delta_+^{\max}(i)}{\Delta_+^{\max}(i)+\Delta_-^{\max}(i)}(F(O^{i-1}) - F(O^{i-1} \cup i)) & \text{if } i \notin OPT \\ \frac{\Delta_-^{\max}(i)}{\Delta_+^{\max}(i)+\Delta_-^{\max}(i)}(F(O^{i-1}) - F(O^{i-1}\backslash i)) & \text{if } i \in OPT \end{cases}$$

$$\leq \begin{cases} \frac{\Delta_+^{\max}(i)}{\Delta_+^{\max}(i)+\Delta_-^{\max}(i)}(F(B^{i-1}\backslash i) - F(B^{i-1})) & \text{if } i \notin OPT \\ \frac{\Delta_-^{\max}(i)}{\Delta_+^{\max}(i)+\Delta_-^{\max}(i)}(F(A^{i-1} \cup i) - F(A^{i-1})) & \text{if } i \in OPT \end{cases}$$

$$= \begin{cases} \frac{\Delta_+^{\max}(i)}{\Delta_+^{\max}(i)+\Delta_-^{\max}(i)}\Delta_-(i) & \text{if } i \notin OPT \\ \frac{\Delta_-^{\max}(i)}{\Delta_+^{\max}(i)+\Delta_-^{\max}(i)}\Delta_+(i) & \text{if } i \in OPT \end{cases}$$

$$\leq \begin{cases} \frac{\Delta_+^{\max}(i)}{\Delta_+^{\max}(i)+\Delta_-^{\max}(i)}\Delta_-^{\max}(i) & \text{if } i \notin OPT \\ \frac{\Delta_-^{\max}(i)}{\Delta_+^{\max}(i)+\Delta_-^{\max}(i)}\Delta_+^{\max}(i) & \text{if } i \in OPT \end{cases}$$

$$= \frac{\Delta_+^{\max}(i)\Delta_-^{\max}(i)}{\Delta_+^{\max}(i) + \Delta_-^{\max}(i)}$$

where the first inequality is due to submodularity: $O^{i-1}\backslash i \supseteq A^{i-1}$ and $O^{i-1} \cup i \subseteq B^{i-1}$.

Putting the above inequalities together:

$$E\left[F(O^{i-1}) - F(O^i) - \frac{1}{2}\left(F(A^i) - F(A^{i-1}) + F(B^i) - F(B^{i-1}) + \rho_i\right)\Big|A^{i-1}, \widehat{A}_i, \widehat{B}_i\right]$$

$$\leq \frac{1/2}{\Delta_+^{\max}(i) + \Delta_-^{\max}(i)}\left[2\Delta_+^{\max}(i)\Delta_-^{\max}(i) - \Delta_-^{\max}(i)(\Delta_-^{\max}(i) - \rho_i)\right.$$

$$\left. - \Delta_+^{\max}(i)(\Delta_+^{\max}(i) - \rho_i)\right] - \frac{1}{2}\rho_i$$

$$= \frac{1/2}{\Delta_+^{\max}(i) + \Delta_-^{\max}(i)}\left[-(\Delta_+^{\max}(i) - \Delta_-^{\max}(i))^2 + \rho_i(\Delta_+^{\max}(i) + \Delta_-^{\max}(i))\right] - \frac{1}{2}\rho_i$$

$$\leq \frac{\frac{1}{2}\rho_i(\Delta_+^{\max}(i) + \Delta_-^{\max}(i))}{\Delta_+^{\max}(i) + \Delta_-^{\max}(i)} - \frac{1}{2}\rho_i$$

$$= 0.$$

**Case 2:** $0 < \Delta_+(i) \leq \Delta_+^{\max}(i)$, $\Delta_-^{\max}(i) < 0$. In this case, the algorithm always choses to include $i$, so $A^i = A^{i-1} \cup i$, $B^i = B^{i-1}$ and $O^i = O^{i-1} \cup i$:

$$E[F(A^i) - F(A^{i-1})|A^{i-1}, \widehat{A}_i, \widehat{B}_i] = F(A^{i-1} \cup i) - F(A^{i-1}) = \Delta_+(i) > 0$$

$$E[F(B^i) - F(B^{i-1})|A^{i-1}, \widehat{A}_i, \widehat{B}_i] = F(B^{i-1}) - F(B^{i-1}) = 0$$

$$E[F(O^{i-1}) - F(O^i)|A^{i-1}, \widehat{A}_i, \widehat{B}_i] = F(O^{i-1}) - F(O^{i-1} \cup i)$$

$$\leq \begin{cases} 0 & \text{if } i \in OPT \\ F(B^{i-1}\backslash i) - F(B^{i-1}) & \text{if } i \notin OPT \end{cases}$$

$$= \begin{cases} 0 & \text{if } i \in OPT \\ \Delta_-(i) & \text{if } i \notin OPT \end{cases}$$

$$\leq 0$$

$$< \frac{1}{2}E[F(A^i) - F(A^{i-1}) + F(B^i) - F(B^{i-1}) + \rho_i|A^{i-1}, \widehat{A}_i, \widehat{B}_i]$$

where the first inequality is due to submodularity: $O^{i-1} \cup i \subseteq B^{i-1}$.

**Case 3:** $\Delta_+(i) \leq 0 < \Delta_+^{\max}(i), 0 < \Delta_-(i) < \Delta_-^{\max}(i)$. Analogous to Case 1.

**Case 4:** $\Delta_+(i) \leq 0 < \Delta_+^{\max}(i), \Delta_-(i) \leq 0$. This is not possible, by Lemma C.1.

**Case 5:** $\Delta_+(i) \leq \Delta_+^{\max}(i) \leq 0, 0 < \Delta_-(i) \leq \Delta_-^{\max}(i)$. Analogous to Case 2.

**Case 6:** $\Delta_+(i) \leq \Delta_+^{\max}(i) \leq 0, \Delta_-(i) \leq 0$. This is not possible, by Lemma C.1.

$\square$

We will now prove the main theorem.

**Theorem 6.1.** *Let $F$ be a non-negative submodular function. CF-2g solves the unconstrained problem $\max_{A \subset V} F(A)$ with worst-case approximation factor $E[F(A_{CF})] \geq \frac{1}{2}F^* - \frac{1}{4}\sum_{i=1}^{N} E[\rho_i]$, where $A_{CF}$ is the output of the algorithm, $F^*$ is the optimal value, and $\rho_i = \max\{\Delta_+^{\max}(e) - \Delta_+(e), \Delta_-^{\max}(e) - \Delta_-(e)\}$ is the maximum discrepancy in the marginal gain due to the bounds.*

*Proof.* Summing up the statement of Lemma C.2 for all $i$ gives us a telescoping sum, which reduces to:

$$E[F(O^0) - F(O^n)] \leq \frac{1}{2}E[F(A^n) - F(A^0) + F(B^n) - F(B^0)] + \frac{1}{2}\sum_{i=1}^{n} E[\rho_i]$$

$$\leq \frac{1}{2}E[F(A^n) + F(B^n)] + \frac{1}{2}\sum_{i=1}^{n} E[\rho_i].$$

Note that $O^0 = OPT$ and $O^n = A^n = B^n$, so $E[F(A^n)] \geq \frac{1}{2}F^* - \frac{1}{4}\sum_i E[\rho_i]$. $\square$

## C.1 Example: max graph cut

Let $C_i = (A^{i-1}\backslash\widehat{A}_i) \cup (\widehat{B}_i\backslash B^{i-1})$ be the set of elements concurrently processed with $i$ but ordered after $i$, and $D_i = B^i\backslash A^i$ be the set of elements ordered after $i$. Denote $\bar{A}_i = V\backslash(\widehat{A}_i \cup C_i \cup D_i) = \{1,\ldots,i\}\backslash\widehat{A}_i$ be the elements up to $i$ that are not included in $\widehat{A}_i$. Let $w_i(S) = \sum_{j\in S,(i,j)\in E} w(i,j)$. For the max graph cut function, it is easy to see that

$$\Delta_+ \geq -w_i(\widehat{A}_i) - w_i(C_i) + w_i(D_i) + w_i(\bar{A}_i)$$
$$\Delta_+^{\max} = -w_i(\widehat{A}_i) + w_i(C_i) + w_i(D_i) + w_i(\bar{A}_i)$$
$$\Delta_- \geq +w_i(\widehat{A}_i) - w_i(C_i) + w_i(D_i) - w_i(\bar{A}_i)$$
$$\Delta_-^{\max} = +w_i(\widehat{A}_i) + w_i(C_i) + w_i(D_i) - w_i(\bar{A}_i)$$

Thus, we can see that $\rho_i \leq 2w_i(C_i)$.

Suppose we have bounded delay $\tau$, so $|C_i| \leq \tau$. Then $w_i(C_i)$ has a hypergeometric distribution with mean $\frac{\deg(i)}{N}\tau$, and $E[\rho_i] \leq 2\tau\frac{\deg(i)}{N}$. The approximation of the hogwild algorithm is then $E[F(A^n)] \geq \frac{1}{2}F^* - \tau\frac{\#\text{edges}}{2N}$. In sparse graphs, the hogwild algorithm is off by a small additional term, which albeit grows linearly in $\tau$. In a complete graph, $F^* = \frac{1}{2}\#\text{edges}$, so $E[F(A^n)] \geq F^*\left(\frac{1}{2} - \frac{\tau}{N}\right)$, which makes it possible to scale $\tau$ linearly with $N$ while retaining the same approximation factor.

## C.2 Example: set cover

Consider the simple set cover function, for $\lambda < L/N$:

$$F(A) = \sum_{l=1}^{L} \min(1, |A \cap S_l|) - \lambda|A| = |\{l : A \cap S_l \neq \emptyset\}| - \lambda|A|.$$

We assume that there is some bounded delay $\tau$.

Suppose also that the sets $S_l$ form a partition, so each element $e$ belongs to exactly one set. Let $n_l = |S_l|$ denote the size of $S_l$. Given any ordering $\pi$, let $e_l^t$ be the $t$th element of $S_l$ in the ordering, i.e. $|\{e' : \pi(e') \le \pi(e_l^t) \wedge e' \in S_l\}| = t$.

For any $e \in S_l$, we get

$$\Delta_+(e) = -\lambda + 1\{A^{\iota(e)-1} \cap S_l = \emptyset\}$$
$$\Delta_+^{\max}(e) = -\lambda + 1\{\widehat{A}_e \cap S_l = \emptyset\}$$
$$\Delta_-(e) = +\lambda - 1\{B^{\iota(e)-1}\backslash e \cap S_l = \emptyset\}$$
$$\Delta_-^{\max}(e) = +\lambda - 1\{\widehat{B}_e\backslash e \cap S_l = \emptyset\}$$

Let $\eta$ be the position of the first element of $S_l$ to be accepted, i.e. $\eta = \min\{t : e_l^t \in A \cap S_l\}$. (For convenience, we set $\eta = n_l$ if $A \cap S_l = \emptyset$.) We first show that $\eta$ is independent of $\pi$: for $\eta < n_l$,

$$P(\eta|\pi) = \frac{\Delta_+^{\max}(e_l^\eta)}{\Delta_+^{\max}(e_l^\eta) + \Delta_-^{\max}(e_l^\eta)} \prod_{t=1}^{\eta-1} \frac{\Delta_-^{\max}(e_l^t)}{\Delta_+^{\max}(e_l^t) + \Delta_-^{\max}(e_l^t)}$$
$$= \frac{1-\lambda}{1-\lambda+\lambda} \prod_{t=1}^{\eta-1} \frac{\lambda}{1-\lambda+\lambda}$$
$$= (1-\lambda)\lambda^{\eta-1},$$

and $P(\eta = n_l|\pi) = \lambda^{\eta-1}$.

Note that, $\Delta_-^{\max}(e) - \Delta_-(e) = 1$ iff $e = e_l^{n_l}$ is the last element of $S_l$ in the ordering, there are no elements accepted up to $\widehat{B}_{e_l^{n_l}}\backslash e_l^{n_l}$, and there is some element $e'$ in $\widehat{B}_{e_l^{n_l}}\backslash e_l^{n_l}$ that is rejected and not in $B^{\iota(e_l^{n_l})-1}$. Denote by $m_l \le \min(\tau, n_l - 1)$ the number of elements before $e_l^{n_l}$ that are inconsistent between $\widehat{B}_{e_l^{n_l}}$ and $B^{\iota(e_l^{n_l})-1}$. Then $\mathbb{E}[\Delta_-^{\max}(e_l^{n_l}) - \Delta_-(e_l^{n_l})] = P(\Delta_-^{\max}(e_l^{n_l}) \ne \Delta_-(e_l^{n_l}))$ is

$$\lambda^{n_l-1-m_l}(1-\lambda^{m_l}) \quad = \quad \lambda^{n_l-1}(\lambda^{-m_l}-1) \quad \le \quad \lambda^{n_l-1}(\lambda^{-\min(\tau, n_l-1)}-1) \quad \le \quad 1-\lambda^\tau.$$

If $\lambda = 1$, $\Delta_+^{\max}(e) \le 0$, so no elements before $e_l^{n_l}$ will be accepted, and $\Delta_-^{\max}(e_l^{n_l}) = \Delta_-(e_l^{n_l})$.

On the other hand, $\Delta_+^{\max}(e) - \Delta_+(e) = 1$ iff $(A^{\iota(e)-1}\backslash\widehat{A}_e) \cap S_l \ne \emptyset$, that is, if an element has been accepted in $A$ but not yet observed in $\widehat{A}_e$. Since we assume a bounded delay, only the first $\tau$ elements after the first acceptance of an $e \in S_l$ may be affected.

$$\mathbb{E}\left[\sum_{e \in S_l} \Delta_+^{\max}(e) - \Delta_+(e)\right]$$
$$= \mathbb{E}[\#\{e : e \in S_l \wedge e_l^\eta \in A^{\iota(e)-1} \wedge e_l^\eta \notin \widehat{A}_e\}]$$
$$= \mathbb{E}[\mathbb{E}[\#\{e : e \in S_l \wedge e_l^\eta \in A^{\iota(e)-1} \wedge e_l^\eta \notin \widehat{A}_e\} \mid \eta = t, \pi(e_l^t) = k]]$$
$$= \sum_{t=1}^{n_l} \sum_{k=t}^{N-n+t} P(\eta = t, \pi(e_l^t) = k)\mathbb{E}[\#\{e : e \in S_l \wedge e_l^\eta \in A^{\iota(e)-1} \wedge e_l^\eta \notin \widehat{A}_e\} \mid \eta = t, \pi(e_l^t) = k]$$
$$= \sum_{t=1}^{n_l} P(\eta = t) \sum_{k=t}^{N-n+t} P(\pi(e_l^t) = k)\mathbb{E}[\#\{e : e \in S_l \wedge e_l^\eta \in A^{\iota(e)-1} \wedge e_l^\eta \notin \widehat{A}_e\} \mid \eta = t, \pi(e_l^t) = k].$$

Under the assumption that every ordering $\pi$ is equally likely, and a bounded delay $\tau$, conditioned on $\eta = t, \pi(e_l^t) = k$, the random variable $\#\{e : e \in S_l \wedge e_l^\eta \in A^{\iota(e)-1} \wedge e_l^\eta \notin \widehat{A}_e\}$ has hypergeometric distribution with mean $\frac{n_l-t}{N-k}\tau$. Also, $P(\pi(e_l^t) = k) = \frac{n_l}{N}\binom{n-1}{t-1}\binom{N-n}{k-t}/\binom{N-1}{k-1}$, so

the above expression becomes

$$
\mathbb{E}\left[\sum_{e \in S_l} \Delta_+^{\max}(e) - \Delta_+(e)\right]
$$

$$
= \sum_{t=1}^{n_l} P(\eta = t) \sum_{k=t}^{N-n+t} \frac{n_l}{N} \frac{\binom{n-1}{t-1}\binom{N-n}{k-t}}{\binom{N-1}{k-1}} \frac{n-t}{N-k} \tau
$$

$$
= \frac{n_l}{N}\tau \sum_{t=1}^{n_l} P(\eta = t) \sum_{k=t}^{N-n+t} \frac{\binom{k-1}{t-1}\binom{N-k}{n-t}}{\binom{N-1}{n-1}} \frac{n-t}{N-k} \qquad \text{(symmetry of hypergeometric)}
$$

$$
= \frac{n_l}{N}\tau \sum_{t=1}^{n_l} \frac{P(\eta = t)}{\binom{N-1}{n-1}} \sum_{k=t}^{N-n+t} \binom{k-1}{t-1}\binom{N-k-1}{n-t-1}
$$

$$
= \frac{n_l}{N}\tau \sum_{t=1}^{n_l} \frac{P(\eta = t)}{\binom{N-1}{n-1}} \binom{N-1}{n-1} \qquad \text{(Lemma E.1, } a = N-2, b = n_l - 2, j = 1\text{)}
$$

$$
= \frac{n_l}{N}\tau \sum_{t=1}^{n_l} P(\eta = t)
$$

$$
= \frac{n_l}{N}\tau.
$$

Since $\Delta_+^{\max}(e) \geq \Delta_+(e)$ and $\Delta_-^{\max}(e) \geq \Delta_-^{\max}(e)$, we have that $\rho_e \leq \Delta_+^{\max}(e) - \Delta_+(e) + \Delta_-^{\max}(e) - \Delta_-(e)$, so

$$
\mathbb{E}\left[\sum_e \rho_e\right] = \mathbb{E}\left[\sum_e \Delta_+^{\max}(e) - \Delta_+(e) + \Delta_-^{\max}(e) - \Delta_-(e)\right]
$$

$$
= \sum_l \mathbb{E}\left[\sum_{e \in S_l} \Delta_+^{\max}(e) - \Delta_+(e)\right] + \mathbb{E}\left[\sum_{e \in S_l} \Delta_-^{\max}(e) - \Delta_-(e)\right]
$$

$$
\leq \tau \frac{\sum_l n_l}{N} + L(1 - \lambda^\tau)
$$

$$
= \tau + L(1 - \lambda^\tau).
$$

Note that $\mathbb{E}\left[\sum_e \rho_e\right]$ does not depend on $N$ and is linear in $\tau$. Also, if $\tau = 0$ in the sequential case, we get $\mathbb{E}\left[\sum_e \rho_e\right] \leq 0$.

# D  Upper bound on expected number of failed transactions

Let $N$ be the number of elements, i.e. the cardinality of the ground set. Let $C_i = (A^{i-1} \backslash \widehat{A}_i) \cup (\widehat{B}_i \backslash B^{i-1})$. We assume a bounded delay $\tau$, so that $|C_i| \leq \tau$ for all $i$.

We call element $i$ *dependent* on $i'$ if $\exists A, F(A \cup i) - F(A) \neq F(A \cup i' \cup i) - F(A \cup i')$ or $\exists B, F(B \backslash i) - F(B) \neq F(B \cup i' \backslash i) - F(B \cup i')$, i.e. the result of the processing $i'$ will affect the computation of $\Delta$'s for $i$. For example, for the graph cut problem, every vertex is dependent on its neighbors; for the separable sums problem, $i$ is dependent on $\{i' : \exists S_l, i \in S_l, i' \in S_l\}$.

Let $n_i$ be the number of elements that $i$ is dependent on. Now, we note that if $C_i$ does not contain any elements on which $i$ is dependent, then $\Delta_+^{\max}(i) = \Delta_+(i) = \Delta_+^{\min}(i)$ and $\Delta_-^{\max}(i) = \Delta_-(i) = \Delta_-^{\min}(i)$, so $i$ will not fail. Conversely, if $i$ fails, there must be some element $i' \in C_i$ such that $i$ is dependent on $i'$.

$$
\begin{aligned}
E(\text{number of failed transactions}) &= \sum_i P(i \text{ fails}) \\
&\leq \sum_i P(\exists i' \in C_i, i \text{ depends on } i') \\
&\leq \sum_i E\left[\sum_{i' \in C_i} 1\{i \text{ depends on } i'\}\right] \\
&\leq \sum_i \frac{\tau n_i}{N}
\end{aligned}
$$

The last inequality follows from the fact that $\sum_{i' \in C_i} 1\{i \text{ depends on } i'\}$ is a hypergeometric random variable and $|C_i| \leq \tau$.

Note that the bound established above is generic to functions $F$, and additional knowledge of $F$ can lead to better analyses on the algorithm's concurrency.

## D.1  Upper bound for max graph cut

By applying the above generic bound, we see that the number of failed transactions for max graph cut is upper bounded by $\frac{\tau}{N} \sum_i n_i = \tau \frac{2\#\text{edges}}{N}$.

## D.2  Upper bound for set cover

For the set cover problem, we can provide a tighter bound on the number of failed items. We make the same assumptions as before in the CF-2g analysis, i.e. the sets $S_l$ form a partition of $V$, there is a bounded delay $\tau$.

Observe that for any $e \in S_l$, $\Delta_-^{\min}(e) \neq \Delta_-^{\max}(e)$ if $\widehat{B}_e \backslash e \cap S_l \neq \emptyset$ and $\widetilde{B}_e \backslash e \cap S_l = \emptyset$. This is only possible if $e_l^{n_l} \notin \widetilde{B}_e$ and $\widetilde{B}_e \supset \widehat{A}_e \cap S_l = \emptyset$, that is $\pi(e) \geq \pi(e_l^{n_l}) - \tau$ and $\forall e' \in S_l, (\pi(e') < \pi(e_l^{n_l}) - \tau) \implies (e' \notin A)$. The latter condition is achieved with probability $\lambda^{n_l - m_l}$, where

$m_l = \#\{e' : \pi(e') \geq \pi(e_l^{n_l}) - \tau\}$. Thus,

$$\mathbb{E}\left[\#\{e : \Delta_-^{\min}(e) \neq \Delta_-^{\max}(e)\}\right] = \mathbb{E}[m_l \, 1(\forall e' \in S_l, (\pi(e') < \pi(e_l^{n_l}) - \tau) \implies (e' \notin A))]$$
$$= \mathbb{E}[\mathbb{E}[m_l \, 1(\forall e' \in S_l, (\pi(e') < \pi(e_l^{n_l}) - \tau) \implies (e' \notin A))|u_{1:N}]]$$
$$= \mathbb{E}[m_l \, \mathbb{E}[1(\forall e' \in S_l, (\pi(e') < \pi(e_l^{n_l}) - \tau) \implies (e' \notin A))|u_{1:N}]]$$
$$= \mathbb{E}[m_l \lambda^{n_l - m_l}]$$
$$\leq \lambda^{(n_l - \tau)_+} \mathbb{E}[m_l]$$
$$= \lambda^{(n_l - \tau)_+} \mathbb{E}[\mathbb{E}[m_l|\pi(e_l^{n_l}) = k]]$$
$$= \lambda^{(n_l - \tau)_+} \sum_{k=n_l}^{N} P(\pi(e_l^{n_l}) = k)\mathbb{E}[m_l|\pi(e_l^{n_l}) = k]].$$

Conditioned on $\pi(e_l^{n_l}) = k$, $m_l$ is a hypergeometric random variable with mean $\frac{n_l - 1}{k - 1}\tau$. Also $P(\pi(e_l^{n_l}) = k) = \frac{n_l}{N}\binom{n_l - 1}{0}\binom{N - n_l}{N - k}/\binom{N - 1}{N - k}$. The above expression is therefore

$$\mathbb{E}\left[\#\{e : \Delta_-^{\min}(e) \neq \Delta_-^{\max}(e)\}\right]$$
$$= \lambda^{(n_l - \tau)_+} \sum_{k=n_l}^{N} \frac{n_l}{N} \frac{\binom{n_l - 1}{0}\binom{N - n_l}{N - k}}{\binom{N - 1}{N - k}} \frac{n_l - 1}{k - 1}\tau$$
$$= \lambda^{(n_l - \tau)_+} \frac{n_l}{N}\tau \sum_{k=n_l}^{N} \frac{\binom{N - k}{0}\binom{k - 1}{n_l - 1}}{\binom{N - 1}{n_l - 1}} \frac{n_l - 1}{k - 1} \qquad \text{(symmetry of hypergeometric)}$$
$$= \lambda^{(n_l - \tau)_+} \frac{n_l}{N}\frac{\tau}{\binom{N - 1}{n_l - 1}} \sum_{k=n_l}^{N} \binom{N - k}{0}\binom{k - 2}{n_l - 2}$$
$$= \lambda^{(n_l - \tau)_+} \frac{n_l}{N}\frac{\tau}{\binom{N - 1}{n_l - 1}} \binom{N - 1}{n_l - 1} \qquad \text{(Lemma E.1, } a = N - 2, b = n_l - 2, j = 2, t = n_l\text{)}$$
$$= \lambda^{(n_l - \tau)_+} \frac{n_l}{N}\tau.$$

Now we consider any element $e \in S_l$ with $\pi(e) < \pi(e_l^{n_l}) - \tau$ that fails. (Note that $e_l^{n_l} \in \widehat{B}_e$ and $\widetilde{B}_e$, so $\Delta_-^{\min}(e) = \Delta_-^{\max}(e) = \lambda$.) It must be the case that $\widehat{A}_e \cap S_l = \emptyset$, for otherwise $\Delta_+^{\min}(e) = \Delta_+^{\max}(e) = -\lambda$ and it does not fail. This implies that $\Delta_+^{\max}(e) = 1 - \lambda \geq u_i$. At commit, if $A^{\iota(e) - 1} \cap S_l = \emptyset$, we accept $e$ into $A$. Otherwise, $A^{\iota(e) - 1} \cap S_l \neq \emptyset$, which implies that some other element $e' \in S_l$ has been accepted. Thus, we conclude that every element $e \in S_l$ that fails must be within $\tau$ of the first accepted element $e_l^\eta \, in S_l$. The expected number of such elements is exactly as we computed in the CF-2ganalysis: $\frac{n_l}{N}\tau$.

Hence, the expected number of elements that fails is upper bounded as

$$\mathbb{E}[\#\text{failed transactions}] \leq \sum_l (1 + \lambda^{(n_l - \tau)_+})\frac{n_l}{N}\tau$$
$$\leq \sum_l 2\frac{n_l}{N}\tau$$
$$= 2\tau.$$

# E   Lemma

**Lemma E.1.** $\sum_{k=t}^{a-b+t} \binom{k-j}{t-j}\binom{a-k+j}{b-t+j} = \binom{a+1}{b+1}$.

*Proof.*

$$\sum_{k=t}^{a-b+t} \binom{k-j}{t-j}\binom{a-k+j}{b-t+j}$$

$$= \sum_{k'=0}^{a-b} \binom{k'+t-j}{t-j}\binom{a-k'-t+j}{b-t+j}$$

$$= \sum_{k'=0}^{a-b} \binom{k'+t-j}{k'}\binom{a-k'-t+j}{a-b-k'} \qquad \text{(symmetry of binomial coeff.)}$$

$$= (-1)^{a-b}\sum_{k'=0}^{a-b} \binom{-t+j-1}{k'}\binom{-b+t-j-1}{a-b-k'} \qquad \text{(upper negation)}$$

$$= (-1)^{a-b}\binom{-b-2}{a-b} \qquad \text{(Chu-Vandermonde's identity)}$$

$$= \binom{a+1}{a-b} \qquad \text{(upper negation)}$$

$$= \binom{a+1}{b+1} \qquad \text{(symmetry of binomial coeff.)}$$

$\square$

# F    Parallel algorithms for separable sums

For some functions $F$, we can maintain sketches / statistics to aid the computation of $\Delta_+^{\max}$, $\Delta_-^{\max}$, $\Delta_+^{\min}$, $\Delta_-^{\min}$. In particular, we consider functions of the form $F(X) = \sum_{l=1}^{L} g\left(\sum_{i \in X \cup S_l} w_l(i)\right) - \lambda \sum_{i \in X} v(i)$, where $S_l \subseteq V$ are (possibly overlapping) groups of elements in the ground set, $g$ is a non-decreasing concave scalar function, and $w_l(i)$ and $v(i)$ are non-negative scalar weights. An example of such functions is set cover $F(A) = \sum_{l=1}^{L} \min(1, |A \cup S_l|) - \lambda |A|$. It is easy to see that $F(X \cup e) - F(X) = \sum_{l: e \in S_l} \left[ g\left(w_l(e) + \sum_{i \in X \cup S_l} w_l(i)\right) - g\left(\sum_{i \in X \cup S_l} w_l(i)\right) \right] - \lambda v(e)$. Define

$$\widehat{\alpha}_l = \sum_{j \in \widehat{A} \cup S_l} w_l(j), \qquad \widehat{\alpha}_{l,e} = \sum_{j \in \widehat{A}_e \cup S_l} w_l(j), \qquad \alpha_l^{\iota(e)-1} = \sum_{j \in A^{\iota(e)-1} \cup S_l} w_l(j).$$

$$\widehat{\beta}_l = \sum_{j \in \widehat{B} \cup S_l} w_l(j), \qquad \widehat{\beta}_{l,e} = \sum_{j \in \widehat{B}_e \cup S_l} w_l(j), \qquad \beta_l^{\iota(e)-1} = \sum_{j \in B^{\iota(e)-1} \cup S_l} w_l(j).$$

## F.1    CF-2g for separable sums $F$

Algorithm 9 updates $\widehat{\alpha}_l$ and $\widehat{\beta}_l$, and computes $\Delta_+^{\max}(e)$ and $\Delta_-^{\max}(e)$ using $\widehat{\alpha}_{l,e}$ and $\widehat{\beta}_{l,e}$. Following arguments analogous to that of Lemma 4.1, we can show:

**Lemma F.1.** *For each $l$ and $e \in V$, $\widehat{\alpha}_{l,e} \leq \alpha_l^{\iota(e)-1}$ and $\widehat{\beta}_{l,e} \geq \beta_l^{\iota(e)-1}$.*

**Corollary F.2.** *Concavity of $g$ implies that $\Delta$'s computed by Algorithm 9 satisfy*

$$\Delta_+^{\max}(e) \quad \geq \quad \sum_{S_l \ni e} \left[ g(\alpha_l^{\iota(e)-1} + w_l(e)) - g(\alpha_l^{\iota(e)-1}) \right] - \lambda v(e) \quad = \quad \Delta_+(e),$$

$$\Delta_-^{\max}(e) \quad \geq \quad \sum_{S_l \ni e} \left[ g(\beta_l^{\iota(e)-1} - w_l(e)) - g(\beta_l^{\iota(e)-1}) \right] + \lambda v(e) \quad = \quad \Delta_-(e),$$

The analysis of Section 6.1 follows immediately from the above.

---

**Algorithm 9:** CF-2g for separable sums

1  **for** $e \in V$ **do** $\widehat{A}(e) = 0$

3  **for** $l = 1, \dots, L$ **do** $\widehat{\alpha}_l = 0$, $\widehat{\beta}_l = \sum_{e \in S_l} w_l(e)$

5  **for** $p \in \{1, \dots, P\}$ **do in parallel**
6      **while** $\exists$ *element to process* **do**
7        $e$ = next element to process
8        $\Delta_+^{\max}(e) = -\lambda v(e) + \sum_{S_l \ni e} g(\widehat{\alpha}_l + w_l(e)) - g(\widehat{\alpha}_l)$
9        $\Delta_-^{\max}(e) = +\lambda v(e) + \sum_{S_l \ni e} g(\widehat{\beta}_l - w_l(e)) - g(\widehat{\beta}_l)$
10       Draw $u_e \sim Unif(0, 1)$
11       **if** $u_e < \frac{[\Delta_+^{\max}(e)]_+}{[\Delta_+^{\min}(e)]_+ + [\Delta_-^{\max}(e)]_+}$ **then**
12         $\widehat{A}(e) \leftarrow 1$
13         **for** $l : e \in S_l$ **do**
14           $\widehat{\alpha}_l \leftarrow \widehat{\alpha}_l + w_l(e)$
15       **else**
16         **for** $l : e \in S_l$ **do**
17           $\widehat{\beta}_l \leftarrow \widehat{\beta}_l - w_l(e)$

---

## F.2  CC-2g for separable sums $F$

Analogous to the CF-2g algorithm, we maintain $\widehat{\alpha}_l, \widehat{\beta}_l$ and additionally $\widetilde{\alpha}_l = \sum_{j \in \widetilde{A} \cup S_l} w_l(j)$ and $\widetilde{\beta}_l = \sum_{j \in \widetilde{B} \cup S_l} w_l(j)$. Following the arguments of Lemma 5.1 and Corollary 5.3, we can show the following.

**Lemma F.3.** $\widehat{\alpha}_{l,e} \le \alpha^{\iota(e)-1} \le \widetilde{\alpha}_{l,e} - w_l(e)$ and $\widehat{\beta}_{l,e} \ge \beta^{\iota(e)-1} \ge \widetilde{\beta}_{l,e} + w_l(e)$

**Corollary F.4.** *Concavity of $g$ implies that the $\Delta$'s computed by Algorithm 10 satisfy:*

$$\Delta_+^{\max}(e) = -\lambda v(e) + \sum_{S_l \ni e} [g(\widehat{\alpha}_{l,e} + w_l(e)) - g(\widehat{\alpha}_{l,e})]$$

$$\ge -\lambda v(e) + \sum_{S_l \ni e} \left[ g(\widehat{\alpha}_l^{\iota(e)-1} + w_l(e)) - g(\widehat{\alpha}_l^{\iota(e)-1}) \right] \qquad = \Delta_+(e)$$

$$\ge -\lambda v(e) + \sum_{S_l \ni e} [g(\widetilde{\alpha}_{l,e}) - g(\widetilde{\alpha}_{l,e} - w_l(e))] \qquad = \Delta_+^{\min}(e),$$

$$\Delta_-^{\max}(e) = \lambda v(e) + \sum_{S_l \ni e} \left[ g(\widehat{\beta}_{l,e} - w_l(e)) - g(\widehat{\beta}_{l,e}) \right]$$

$$\ge \lambda v(e) + \sum_{S_l \ni e} \left[ g(\widehat{\beta}_l^{\iota(e)-1} - w_l(e)) - g(\widehat{\beta}_l^{\iota(e)-1}) \right] \qquad = \Delta_-(e)$$

$$\ge \lambda v(e) + \sum_{S_l \ni e} \left[ g(\widetilde{\beta}_l^{\iota(e)-1}) - g(\widetilde{\beta}_l^{\iota(e)-1} + w_l(e)) \right] \qquad = \Delta_-^{\min}(e).$$

The analysis of Section 6.3 and 6.2 follows immediately from the above.

---

**Algorithm 10:** CC-2g for separable sums

1  **for** $e \in V$ **do** $\widehat{A}(e) = \widetilde{A}(e) = 0, \widehat{B}(e) = \widetilde{B}(e) = 1$
2
3  **for** $l = 1, \ldots, L$ **do**
4       $\widehat{\alpha}_l = \widetilde{\alpha}_l = 0$
5       $\widehat{\beta}_l = \widetilde{\beta}_l = \sum_{e \in S_l} w_l(e)$
6  **for** $i = 1, \ldots, |V|$ **do** processed$(i) = false$
7
8  $\iota = 0$
9  **for** $p \in \{1, \ldots, P\}$ **do in parallel**
10       **while** $\exists$ *element to process* **do**
11           $e$ = next element to process
12           $(\widehat{\alpha}_{\cdot,e}, \widetilde{\alpha}_{\cdot,e}, \widehat{\beta}_{\cdot,e}, \widetilde{\beta}_{\cdot,e})$ = getGuarantee$(e)$
13           (result, $u_e$) = propose$(e, \widehat{\alpha}_{\cdot,e}, \widetilde{\alpha}_{\cdot,e}, \widehat{\beta}_{\cdot,e}, \widetilde{\beta}_{\cdot,e})$
14           commit$(e, i, u_e, \text{result})$

---

**Algorithm 11:** CC-2g getGuarantee$(e)$ for separable sums

1  $\widetilde{A}(e) \leftarrow 1; \widetilde{B}(e) \leftarrow 0$
2  **for** $l : e \in S_l$ **do**
3       $\widetilde{\alpha}_l \leftarrow \widetilde{\alpha}_l + w_l(e)$
4       $\widetilde{\beta}_l \leftarrow \widetilde{\beta}_l - w_l(e)$
5  $i = \iota; \iota \leftarrow \iota + 1$
6  $\widehat{\alpha}_{\cdot,e} = \widehat{\alpha}_\cdot; \widehat{\beta}_{\cdot,e} = \widehat{\beta}_\cdot$
7  $\widetilde{\alpha}_{\cdot,e} = \widetilde{\alpha}_\cdot; \widetilde{\beta}_{\cdot,e} = \widetilde{\beta}_\cdot$
8  **return** $(\widehat{\alpha}_{\cdot,e}, \widetilde{\alpha}_{\cdot,e}, \widehat{\beta}_{\cdot,e}, \widetilde{\beta}_{\cdot,e})$

---

**Algorithm 12:** CC-2g propose$(e, \widehat{\alpha}_{\cdot,e}, \widetilde{\alpha}_{\cdot,e}, \widehat{\beta}_{\cdot,e}, \widetilde{\beta}_{\cdot,e})$ for separable sums

---

1   $\Delta_+^{\min}(e) = -\lambda v(e) + \sum_{S_l \ni e} g(\widetilde{\alpha}_l) - g(\widetilde{\alpha}_l - w_l(e))$

2   $\Delta_+^{\max}(e) = -\lambda v(e) + \sum_{S_l \ni e} g(\widehat{\alpha}_l + w_l(e)) - g(\widehat{\alpha}_l)$

3   $\Delta_-^{\min}(e) = +\lambda v(e) + \sum_{S_l \ni e} g(\widetilde{\beta}_l) - g(\widetilde{\beta}_l + w_l(e))$

4   $\Delta_-^{\max}(e) = +\lambda v(e) + \sum_{S_l \ni e} g(\widehat{\beta}_l - w_l(e)) - g(\widehat{\beta}_l)$

5   Draw $u_e \sim Unif(0,1)$

6   **if** $u_e < \frac{[\Delta_+^{\min}(e)]_+}{[\Delta_+^{\min}(e)]_+ + [\Delta_-^{\max}(e)]_+}$ **then** result $\leftarrow 1$

8   **else if** $u_e > \frac{[\Delta_+^{\max}(e)]_+}{[\Delta_+^{\max}(e)]_+ + [\Delta_-^{\min}(e)]_+}$ **then** result $\leftarrow -1$

10   **else** result $\leftarrow$ FAIL

12   **return** (result, $u_e$)

---

**Algorithm 13:** CC-2g commit$(e, i, u_e, \text{result})$ for separable sums

---

1   **wait until** $\forall j < i$, processed$(j) = true$

2   **if** *result = FAIL* **then**

3      $\Delta_+^{\text{exact}}(e) = -\lambda v(e) + \sum_{S_l \ni e} g(\widehat{\alpha}_l + w_l(e)) - g(\widehat{\alpha}_l)$

4      $\Delta_-^{\text{exact}}(e) = +\lambda v(e) + \sum_{S_l \ni e} g(\widehat{\beta}_l - w_l(e)) - g(\widehat{\beta}_l)$

5      **if** $u_e < \frac{[\Delta_+^{exact}(e)]_+}{[\Delta_+^{exact}(e)]_+ + [\Delta_-^{exact}(e)]_+}$ **then** result $\leftarrow 1$

7      **else** result $\leftarrow -1$

9   **if** *result* $= 1$ **then**

10      $\widehat{A}(e) \leftarrow 1$

11      $\widetilde{B}(e) \leftarrow 1$

12      **for** $l : e \in S_l$ **do**

13         $\widehat{\alpha}_l \leftarrow \widehat{\alpha}_l + w_l(e)$

14         $\widetilde{\beta}_l \leftarrow \widetilde{\beta}_l + w_l(e)$

15   **else**

16      $\widetilde{A}(e) \leftarrow 0$; $\widehat{B}(e) \leftarrow 0$

17      **for** $l : e \in S_l$ **do**

18         $\widetilde{\alpha}_l \leftarrow \widetilde{\alpha}_l - w_l(e)$

19         $\widehat{\beta}_l \leftarrow \widehat{\beta}_l - w_l(e)$

20   processed$(i) = true$

---

# G   Full experiment results

Figure 5: Experimental results on Erdos-Renyi and ZigZag synthetic graphs.

Figure 6: Set cover on 4 real graphs.

Figure 7: Max graph cut on 4 real graphs.

Figure 8: Experimental results for ring graph on set cover problem.

# H   Illustrative examples

The following examples illustrate how (i) the simple (uni-directional) greedy algorithm may fail for non-monotone submodular functions, and (ii) where the coordination-free double greedy algorithm can run into trouble.

## H.1   Greedy and non-monotone functions

For illustration, consider the following toy example of a non-monotone submodular function. We are given a ground set $V = \{v_0, v_1, v_2, \ldots, v_k\}$ of $k + 1$ elements, and a universe $U = \{u_1, \ldots, u_k\}$. Each element $v_i$ in $V$ covers elements $\mathrm{Cov}(v_i) \subseteq U$ of the universe. In addition, each element in $V$ has a cost $c(v_i)$. We are aiming to maximize the submodular function

$$F(S) = \left| \bigcup_{v \in S} \mathrm{Cov}(v) \right| - \sum_{v \in S} c(v). \tag{3}$$

Let the costs and coverings be as follows:

$$\mathrm{Cov}(v_0) = U \qquad c(v_0) = k - 1 \tag{4}$$

$$\mathrm{Cov}(v_i) = u_i \qquad c(v_i) = \epsilon < 1/k^2 \quad \text{for all } i > 0. \tag{5}$$

Then the optimal solution is $S^* = V \setminus v_0$ with $F(S^*) = k - k\epsilon$.

The greedy algorithm of Nemhauser et al. [8] always adds the element with the largest marginal gain. Since $F(v_0) = 1$ and $F(v_i) = 1 - \epsilon$ for all $i > 0$, the algorithm would pick $v_0$ first. After that, any additional element only has a negative marginal gain, $F(\{v_0, v_i\}) - F(v_0) = -\epsilon$. Hence, the algorithm would end up with a solution $F(v_0) = 1$ or worse, which means an approximation factor of only approximately $1/k$.

For the double greedy algorithm, the scenario would be the following. If $v_0$ happens to be the first element, then it is picked with probability

$$P(v_0) = \frac{[F(v_0) - F(\emptyset)]_+}{[F(v_0) - F(\emptyset)]_+ + [F(V \setminus v_0) - F(V)]_-} = \frac{1}{1 + (k - 1)} = \frac{1}{k}. \tag{6}$$

If $v_0$ is selected, nothing else will be added afterwards, since $[F(v_0, v_i) - F(v_0)]_+ = 0$. If it does not pick $v_0$, then any other element is added with a probability of

$$P(v_i \mid \neg v_0) = \frac{[F(v_i) - F(\emptyset)]_+}{[F(v_i) - F(\emptyset)]_+ + F(V \setminus \{v_0, v_i\}) - F(V \setminus v_0)]_-} = \frac{1 - \epsilon}{1 - \epsilon} = 1. \tag{7}$$

If $v_0$ is not the first element, then any element before $v_0$ is added with probability $p(v_i) = 1 - \epsilon$, and as soon as an element $v_i$ has been picked, $v_0$ will not be added any more. Hence, with high probability, this algorithm returns the optimal solution. The deterministic version surely does.

## H.2   Coordination vs no coordination

The following example illustrates the differences between coordination and no coordination. In this example, let $V$ be split into $m$ disjoint groups $G_j$ of equal size $k = |V|/m$, and let

$$F(S) = \sum_{j=1}^{m} \min\{1, |S \cap G_j|\} - \frac{|S \cap G_j|}{k}. \tag{8}$$

A maximizing set $S^*$ contains one element from each group, and $F(S^*) = m - m/k$.

If the sequential double greedy algorithm has not picked an element from a group, it will retain the next element from that group with probability

$$\frac{1 - 1/k}{1 - 1/k + 1/k} = 1 - 1/k. \tag{9}$$

Once it has sampled an element from a group $G_j$, it does not pick any more elements from $G_j$, and therefore $|S \cap G_j| \leq 1$ for all $j$ and the set $S$ returned by the algorithm. The probability that $S$

does not contain any element from $G_j$ is $k^{-k}$ —fairly low. Hence, with probability $1 - m/k^k$ the algorithm returns the optimal solution.

Without coordination, the outcome heavily depends on the order of the elements. For simplicity, assume that $k$ is a multiple of the number $q$ of processors (or $q$ is a multiple of $k$). In the worst case, the elements are sorted by their groups and the members of each group are processed in parallel. With $q$ processors working in parallel, the first $q$ elements from a group $G$ (up to shifts) will be processed with a bound $\widehat{A}$ that does not contain any element from $G$, and will each be selected with probability $1 - 1/k$. Hence, in expectation, $|S \cap G_j| = \min\{q, k\}(1 - 1/k)$ for all $j$.

If $q > k$, then in expectation $k - 1$ elements from each group are selected, which corresponds to an approximation factor of

$$\frac{m(1 - \frac{k-1}{k})}{m(1 - 1/k)} = \frac{1}{k - 1}. \tag{10}$$

If $k > q$, then in expectation we obtain an approximation factor of

$$\frac{m(1 - \frac{q(1-1/k)}{k})}{m(1 - 1/k)} = 1 - \frac{q}{k} + \frac{1}{k - 1} \tag{11}$$

which decreases linearly in $q$. If $q = k$, then the factor is $1/(q - 1)$ instead of $1/2$.