[Reviews · NeurIPS 2014]

Submitted by Assigned_Reviewer_5

This paper is on solving non-monotone submodular maximization problems at scale. In particular, the paper looks at the double-greedy algorithm for unconstrained submodular maximization and offers modifications to parallelize execution of the algorithm on massive data.

Although on the positive side, the paper addresses an interesting problem and proposes a practical modification of the standard algorithm, I was somewhat disappointed with the theoretical analysis. The first approach CF-2g provides no approximation guarantee for the problem, and the second approach CC-2g that preserves the optimality of the serial double greedy algorithm,
might not provide any improvement in terms of the time complexity. This is very important for an algorithm that is designed for extracting information from large data sets.

Moreover, the paper is near-trivial on the theory front. The analysis is so obvious from a theoretical perspective. The guarantees provided in section 6 are limited to two specific problem instances, namely max-cut and set cover problems, and don't hold in general.

Finally, as there might be a high amount of communication between clients and server, the proposed methods do not easily be extended to work in distributed environments with several computers using Hadoop or other standard frameworks for parallel and distributed computing.
Summary: The problem itself is quite motivated. There have been a few earlier work on trying to "speed up" or "parallelize" the sequential algorithms for the submodular maximization problem. However, the theoretical contribution of this paper is very weak and limited to specific problem instances.

Submitted by Assigned_Reviewer_15

In this paper, the authors propose two parallel algorithms for the double-greedy method, which can achieve the (currently-known) best approximation guarantee for non-monotone submodular maximization. I think the overall quality of this paper is high.

Parallel implementations for popular combinatorial algorithms, including greedy methods, have been actively discussed in a variety of communities other than ML. I think the surveys about the existing studies of this paper would not be very thorough. It is biased to submodular maximization. Since the greedy methods, such as the double-greedy method, can be applied to problems other than submodular maximization. However, to the best of my knowledge, this kind of parallelization of the double-greedy method has not been discussed so far.

Another concern is that CC-2g, which has the same approximation guarantee with the serial one, seems to fail empirically (especially, Fig 4 (b)). The scheme of parallelization of CC-2g becomes slower than the serial implementation. Although the authors claim that there is a trade-off (in approximation quality and scalability) between two algorithms, I wonder how we should interpret this point if CC-2g can be slower than the serial one.
Summary: In this paper, the authors propose two parallel algorithms for the double-greedy method, which can achieve the (currently-known) best approximation guarantee for non-monotone submodular maximization. I think the overall quality of this paper is high, but have a few minor concerns on the authors' claims

Submitted by Assigned_Reviewer_22

The paper discusses the parallel version of a known algorithm (double greedy) for maximization of submodular problems.
The paper describe novel and needed improvement: in the world of big data, parallel algorithm that scale are a must.
Correctness proof is a good addition to the experimental results that show the effectiveness of the algorithm on diverse examples of real life data.
Summary: Well written paper that nicely solves an important problem.

Submitted by Assigned_Reviewer_43

Many submodular maximization could be optimized with decent approximation ratio by greedy-like algorithms. However, due to the inherently serial nature of the algorithms, they could not be easily parallelized. Leveraging the transaction processing model from database system, this paper presents two approaches to parallelizing the double greedy algorithm for unconstrained submodular maximization. In particular, the double greedy algorithm maintains two sets A and B where one can only add elements to A or removed elements from B. The authors consider each such operation an transaction just like the one in database system. The problem is then reduced to how to apply these transactions in parallel.

The first approach (CF-2g) this paper proposes is a hogwild-style coordination-free algorithm where each thread runs locally with a possibly stale version of A and B. Due to the submodularity, the marginal improvement computed within each thread is always an upper bound of the true one. The authors also analyze the approximation ratio of CF-2g.

The second approach (CC-2g) is a concurrency-control algorithm which guarantees the serializability and thus it shares the same approximation ratio as the serial double-greedy algorithm. It works by maintaining the upper and lower bound of A and B, which can be used to locally compute the upper bound and lower bound of the threshold in the serial algorithm. This "often" enables each thread to locally decide to add the element to A or remove it from B. And uncertain transactions are recomputed serially by the server. Figure 2 is a nice illustration of this process.

As a consequence CF-2g promotes efficiency and CC-2g promotes solution quality. At the end, the authors conduct comprehensive experiments to evaluate the proposed algorithms. But seems like CC-2g algorithm is comparable with CF-2g in terms of speedup, and it even has more speedup than CF-2g in some cases (e.g. Fig. 3c) which is counter-intuitive. Is that due to some specific problem structure? I think the authors should give more discussions to make it more clear. Also, as pointed out in the last section, the delays might hurt the performance in distributed environment. But so far it's unclear empirically.
Summary: The proposed methods are novel combinations of submodular optimization and parallel database system. This paper is well written and could potentially inspire more work on parallelizing submodular optimization.
Author Feedback
Author rebuttal: We would like to thank the reviewers for their insights and suggestions, and attempt to address some of their concerns below.

Assigned_Reviewer_15

The reviewer notes that in the experiments the CC-2g algorithm can be slower than the sequential algorithm. This does not violate the asymptotic guarantees but is instead due to a constant (factor of 2) overhead associated with the CC-2g protocol. It is important to note that this overhead is parallelizable and therefore vanishes with increasing processing resources.

The remaining overhead of CC-2g is due to conflict resolution and is an essential part of the algorithm. While CC-2g scales well on randomized orderings, we also assessed its performance on adversarial orderings intentionally designed to emphasize this overhead. The focus of the CC-2g approach is in maintaining the strong approximation guarantee, even in such pathological cases, while providing potential scaling, where possible. Conversely, the CF-2g approach focuses on scaling, even in such pathological cases, while providing potentially good approximations, where possible. We also do not rule out the possibility of an interpolation between the 2 approaches.

The reviewer also notes that the double greedy algorithm has been applied to other combinatorial problems. We admit that we are unaware of such applications, and would be delighted to add additional references. Being able to apply our parallel algorithms to a larger class of problems would further strengthen our case. On the other hand, to the best of our knowledge, our paper is the first to propose a parallel double greedy algorithm.

Assigned_Reviewer_22

We are unfortunately unable to address the reviewer’s concerns, despite the relatively low score, since none were raised in the review text.

Assigned_Reviewer_43
The reviewer expressed confusion over the better speedup of CC-2g as compared to CF-2g, especially given that the CF-2g actually has less coordination overhead (at the expense of approximation quality). This is partly a consequence of the way in which speedup is typically calculated -- speedup is measured relative to the running time of the same algorithm on a single processor.
To effectively compare across algorithms it is necessary to examine the runtime plots (Figure 3a) in which we find that CF-2g is still twice as fast the the CC-2g algorithm. Due to the different bases for computing speedup, it is less meaningful to contrast the speedup of CF-2g with that of CC-2g. Rather, the takeaway of Figure 3b, 3c is that both algorithms have good speedup properties, demonstrating an efficient use of the multicore resources.

It was also suggested that the empirical results do not support the hypothesis that delays in distributed environments should affect performance. Indeed, in this paper, we have focused solely on multicore environments (which present their own challenges), and left distributed algorithms as future work. We speculate that the distributed environment poses greater challenges related to managing the evaluation protocol but this is beyond the scope of our paper.

Assigned_Reviewer_5

The reviewer claims that our paper is weak in its theoretical contribution. While the analysis follows naturally from the assumptions, the challenge was in formulating the setting in a thoughtful way that reflects the underlying computation. Our analysis provides several novel insights and important contributions:

(1) We are able to connect the degradation in CC-2g scalability with the degradation in the CF-2g approximation factor via the maximum inter-processor message delay (tau). We realize that we did not emphasize this connection in the text and will address this in the final version.

(2) The results provide theoretical guarantees on the expected solution quality of CF-2g and scalability of CC-2g.

(3) Section 6 shows how to apply our main theorem (Thm 6.1) to two problems with different structure. We believe our results could be applied more broadly beyond these two examples.

The reviewer raises the concern that CC-2g may not scale and therefore does not improve the time complexity as a function of the number of processors. Although the CC-2g algorithm has a worst case asymptotic complexity of O(n), we show that under reasonable assumptions the coordination overhead is minimal enabling strong scaling. Empirically we evaluate both the worst and average case scaling performance, and demonstrate that in the expected case CC-2g does reduce the actual runtime, which is the key metric of interest. (See also response to Assigned_Reviewer_15.)

Furthermore, the CC-2g algorithm is actually asymptotically work efficient in that the introduction of parallel resources does not increase the computation overhead by more than a constant factor.

We acknowledge the reviewer’s concern about extending our approach to distributed environments. However, we feel that this should not be a criticism of this paper, whose focus is on multicore systems which pose a unique set of challenges. Furthermore, we have shown that we can handle large graphs without stretching the limits of multicore systems.